# Variations of Drought Tendency, Frequency, and Characteristics and Their Responses to Climate Change under CMIP5 RCP Scenarios in Huai River Basin, China

**Jingcai Wang** [1],*[iD]**, Hui Lin** [1]**, Jinbai Huang** [1]**, Chenjuan Jiang** [2]**, Yangyang Xie** [1] **and Mingyao Zhou** [1]

[1] Department of Agriculture and Water Resources Engineering, Yangzhou University, Yangzhou 225009, China; LinhuiW@163.com (H.L.); huangjinbai@aliyun.com (J.H.); xieyang_yang_cool@126.com (Y.X.); myzhou@yzu.edu.cn (M.Z.)

[2] Department of Water Conservancy and Port Engineering, Yangzhou University, Yangzhou 225009, China; jiangchenjuan001@163.com

\* Correspondence: wangjingcai@yzu.edu.cn

**Abstract:** Huai River Basin (HRB) is an important food and industrial production area and a frequently drought-affected basin in eastern China. It is necessary to consider the future drought development for reducing the impact of drought disasters. Three global circulation models (GCMs) from Coupled Model Intercomparison Project phase 5 (CMIP5), such as CNRM-CM5 (CNR), HadGEM2-ES (Had) and MIROC5 (MIR), were used to assessment the future drought conditions under two Representative Concentration Pathways (RCPs) scenarios, namely, RCP4.5 and RCP8.5. The standardized precipitation evapotranspiration index (SPEI), statistical method, Mann-Kendall test, and run theory were carried out to study the variations of drought tendency, frequency, and characteristics and their responses to climate change. The research showed that the three CMIP5 models differ in describing the future seasonal and annual variations of precipitation and temperature in the basin and thus lead to the differences in describing drought trends, frequency, and drought characteristics, such as drought severity, drought duration, and drought intensity. However, the drought trend, frequency, and characteristics in the future are more serious than the history. The drought frequency and characteristics tend to be strengthened under the scenario of high concentration of RCP8.5, and the drought trend is larger than that of low concentration of RCP4.5. The lower precipitation and the higher temperature are the main factors affecting the occurrence of drought. All three CMIP5 models show that precipitation would increase in the future, but it could not offset the evapotranspiration loss caused by significant temperature rise. The serious risk of drought in the future is still higher. Considering the uncertainty of climate models for simulation and prediction, attention should be paid to distinguish the effects of different models in the future drought assessment.

**Keywords:** drought; climate change; standardized precipitation evapotranspiration index; mann-kendall test; run theory; CMIP5

---

## 1. Introduction

Drought is recognized as a natural hazard and environmental disaster and it has caused extensive impact during recent decades over the worldwide [1,2], such as the North America [3,4], Europe [5,6], Australia [7], Africa [8,9], and the Asia [10–12]. Drought is usually caused by the scarcity of precipitation,

coupled with water evaporation, and water consumption expenditure. Drought affects both surface and underground water resources and can lead to reduced water supply, water storage of water conservancy projects, social and economic water use, and ecological environment water use [13]. The lack of water will lead to the destruction of crop water balance and the reduction of crop yield or poor harvest [14], resulting in food problems and even starvation. Drought can also affect ecosystems stability [15] and even kill animals due to a lack of adequate drinking water during severe drought seasons. In recent years, especially in the face of increasing population, agricultural expansion, and industrial and economic development, other aspects of water demand continue to increase, drought will have an important impact on all these aspects [9]. Therefore, drought needs to arouse extensive attention by the human society.

Climate change is a universal consensus of the global scientific community and the public. Recent and potential future increases in global temperature are likely to be associated with the hydrologic cycle, which brings the changes to precipitation and thus increases extreme events such as droughts [16]. Many studies have shown that the drought situation may be changed and even more severe in the face of global and regional climate change throughout different global or regional climate models, emission scenarios, time periods, and drought indicators. Dai [17] has found that the global aridity has increased substantially since the 1970s and the climate models project increased aridity in the 21st century over most of Africa, southern Europe, the Middle East, most of the American continent, Australia, and Southeast Asia. Liu et al. [18] revealed that there would be an increasing drought risk if the global warming continues to rise at a high level. Ahmadalipour et al. [9] took an ensemble of 10 regional climate models and a multi-scalar drought index to quantify drought hazard and found that drought risk in Africa is expected to increase in future with varied rates for different models and scenarios. Meanwhile, different regions and basins will inevitably show different characteristics of drought under the influence of climate change because of the differences in geographical location and economic development scale. Calanca [19] revealed that in the future the Alpine region would suffer an increasing drought frequency and a higher drought severity under the climate scenario of a SRES A2 emission pathway. Leng et al. [20] indicated that droughts would become more severe, prolonged, and frequent for 2020–2049 relative to 1971–2000 in vast areas of China. Meanwhile, even in the same region, drought characteristics would be different for choosing the different global or regional climate models and emission scenarios [21] or selecting the different drought indicators [22]. Therefore, it is necessary to continue carrying out in-depth and systematic drought research to different regions or basins in order to lay a foundation for the local scientific disaster prevention and mitigation responses and measures.

China is a frequently drought-affected country in East Asia since the precipitation and temperature changes significantly from one region to another region [20,23]. Huai River Basin (HRB) is a typical large basin in eastern China and it situated in the middle of the North-South climatic transition zone, which is also one of the most important bases for agriculture production in China. Because of its special geographical location and climatic type, drought and flood have changed dramatically and occurred frequently, which have an important impact on the ecological systems, regional food production security, economy, and the society. In recent decades, with the background of global climate change, the hydrothermal conditions in the HRB have also changed. Some studies reveal that a future increase in evapotranspiration would happen and affect the current humid region south of the HRB retreating southward and changing to a sub-humid region [24]. Yang et al. pointed out that the trend of flood and drought events in the HRB was positively related to climate warming with a higher coefficient of determination [25]. Li et al. also found that drought is expected to rise in frequency, duration, severity, and intensity in the HRB from 2010 to 2099 [26]. The previous studies have shown that the watershed water cycle would be altered under the climate change and then influence the drought conditions. However, it should note that climate change is uncertain. The differences in describing future hydrothermal conditions under different climate models will directly affect the future drought management. As an important food and industrial production area in China, it is necessary to consider

the future drought development under different climate change patterns and emission scenarios in order to cope with potential disaster risks.

Climate models are important tools for studying climate change and its impacts on drought. Climate model has the remarkable ability of simulating the time evolution of average climate characteristics and climate variability. Up to now, there are many global climate models and regional climate models (RCMs) for studying climate change mechanism and exploring climate change factors. A series of GCMs can well simulate the average characteristics of large-scale grids, as well as the characteristics of near-surface temperature, high-level atmospheric field, and atmospheric circulation. The basic idea of RCMs is to nest the RCM into the GCM step by step. The GCM provides the initial field and lateral boundary conditions. The global circulation models (GCMs) of the Coupled Model Intercomparison Project phase 5 (CMIP5) climate models were designed to advance the knowledge of climate variability and climate change by the World Climate Research Programme's (WCRP) Working Group on Coupled Modelling (WGCM) [27]. It provided an important scientific basis for detecting the impact of human activities on historical climate change. CMIP5 have been improved in many aspects comparing with previous generations of CMIP3 and CMIP4, such as experimental design, especially in improving physical parameterization and the temporal and spatial resolution. The CMIP5 models contain more comprehensive patterns and higher resolution. A key to CMIP5's usefulness is that all model output conforms to community standards and is placed in an archive that appears to users as a single unified database. This makes analysis of the multi model ensemble nearly as easy as analysis of a single model. So many studies choose CMIP5 models to analyze climate change. Torres et al. [28] chose several GCMs and forcing scenarios from CMIP3 to CMIP5 to identify South America regions where climate change could be more pronounced in a warmer climate. Basheer et al. [29] used four GCMs from the CMIP5 to assess the impacts of climate change on the streamflow in the Dinder River basin and found that the predicted climate change is likely to affect ecosystems positively and promote the ecological restoration for the habitats. Sun et al. [30] evaluated 14 global climate models in the CMIP5 to capture the extreme climate events and the agricultural climate indices over China. Liu et al. [18] studied the risk-based assessment of changes in global drought and the impact of severe drought on populations under CMIP5 warming conditions and finally given some information and advice to reduce the future drought risk and impact. Ma et al. [24] used the CMIP5 to study the trends in the area of arid/humid climate regions of China and identify the regions of arid/humid patterns change over the next 100 years, and found that the humid region would suffered a significant contraction and the arid/humid transition zones may be lead to an expansion. So many previous studies have tested and verified that the CMIP5 models, as important tools on climate change, could well applied to the climate associated impacts research. In this paper, three CMIP5 climate models, such as CNRM-CM5 (CNR), HadGEM2-ES (Had), and MIROC5 (MIR), would be chosen to analyze the drought response to climate change with two Representative Concentration Pathways (RCPs) scenarios, namely, RCP4.5 and RCP8.5.

Quantification of drought is the premise of drought research. Nowadays, a variety of drought indices have been developed [13,31–33] and applied in the fields of meteorology, hydrology, agriculture, and socio-economic. The drought indices include the standardized precipitation index (SPI), the standardized precipitation evapotranspiration index (SPEI), Keetch–Byram drought index (KBDI), surface water supply index (SWSI), the standardized streamflow index (SSI), Palmer drought severity index (PDSI), crop moisture index (CMI), and vegetation condition index (VCI). Different drought indices play an important role in drought analysis, but there are still some shortcomings and limitations for they were consider different physical mechanism and mathematical model [13,34–37]. Some indices lack the effectiveness of spatial and temporal comparisons, such as the PDSI. While, some indices ignore the effect of evapotranspiration changes caused by the temperature rise, such as the SPI [38]. The American Meteorological Society [39] suggests that the time and space processes of supply and demand are the two basic processes that should be included in an objective definition of drought and in the derivation of a drought index. The standardized precipitation evapotranspiration index is one of

such drought indicators. SPEI was proposed by Vicente-Serrano et al. [40,41] and it has the advantages of the sensitivity to water expenditure in forms of evapotranspiration caused by temperature and the characteristics of multi-time scales like the SPI [42,43]. Based on the above advantages, it has been widely used in different countries and regions over the world [8,44–46], showing a good performance on drought analysis. For China, Li et al. [47] compared the SPEI and the SPI for drought analysis and indicated that the SPEI could better monitor the drought conditions than SPI in months with significant increase of temperature. Shi et al. [48] showed that the SPEI values effectively reflected the spatial and temporal pattern of drought occurrence in Henan Province. Zhao et al. [49] indicated that SPEI was suitable for both short- and long-term drought monitoring compared with PDSI and had a good application prospect in China. Various research results have shown that SPEI has the characteristics of simplicity of calculation and consideration of the impact of temperature variability. It is especially suitable for the study of drought under the current global warming.

Given the importance of this issue and the lack of previous studies, the main objectives of this paper are structured as follows: Section 2 provides short descriptions of the study area and the datasets used in the research. This section also presents brief discretions of the used methods. In Section 3, the future climate changes characteristics of precipitation and temperature in the study area are shown by three different CMIP5 climate models under two emission scenarios (RCP4.5 and 8.5). It also studies the changes of drought tendency and frequency and the differences of drought severity, duration, and intensity historically and in the future. At the end of Section 3, the study discusses how the precipitation and temperature influences the historical and future drought situations under the climate change. Lastly, Section 4 offers the conclusion of this study.

## 2. Materials and Methods

### 2.1. Study Area and the Datasets

The location of the HRB and the distribution of ground meteorological stations were shown in Figure 1. Our study focuses on the upper and middle reaches of the HRB (UMHRB). The area of UMHRB is about $16 \times 10^4$ km$^2$, accounting for 59.3% of the whole basin (about $27 \times 10^4$ km$^2$). The average annual precipitation in UMHRB is about 902 mm (varies from 528 mm to 1308 mm), and its distribution is decreasing from south to north, with mountainous areas more than plains and with coastal areas more than inland areas. The annual average temperature ranges from 13.9 to 16.0 °C. The highest monthly average temperature often occurs in July, while the lowest monthly average temperature occurs in January.

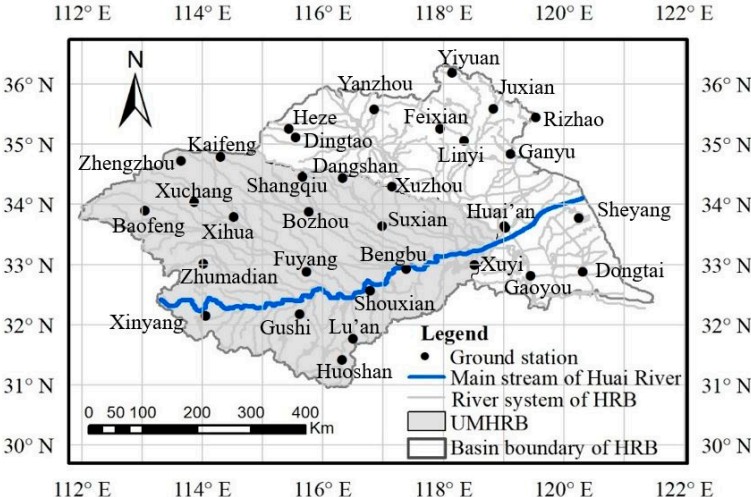

**Figure 1.** Location of the upper and middle reaches of the Huai River Basin (HRB) (UMHRB) and the distribution of ground meteorological stations.

The ground observation datasets derived from the China Meteorological Data Network (CMDN) (http://data.cma.cn/) by the China Meteorological Administration (CMA), which contains monthly precipitation, mean temperature, minimum temperature, maximum temperature, wind speed, and sunshine hours from 1960 to 2015. The missing and some abnormal values have interpolated according to the dataset description documents for quality control. The ground observation data used to test the accuracy of CMIP5 models and calculate the historical SPEI datasets.

The simulated and projected climate datasets come from the CMIP5 models web page (available online at https://esgf-node.llnl.gov/projects/esgf-llnl/). The data file format is NC file and it intercepted based on R software. In this study, the used variables are monthly precipitation and surface air temperature simulated for the present climate (1960–2005) and projected to the 21st century (2020–2099) by some GCMs from CMIP5. Statistical significance tests were conducted to correlate the CMIP5 simulated data with the historical ground observation data during the same periods.

Six CMIP5 models had been analyzed in the previous studies including the CCSM4, CMCC-CM, CNRM-CM5, HadGEM2-ES, MIROC4h, and MIROC5. The basic information of the six CMIP5 models and the accuracy test results were shown in Table 1. The Pearson correlation coefficient (*r*) equal to 1.0 and the mean relative error (*Rmean*) to 0 would indicate a perfect linear relationship between CMIP5 and ground observed data. The Pearson correlation coefficient in Table 1 were statistically significant and had passed the *t* test at *P* = 0.05 level.

**Table 1.** Basic information of Coupled Model Intercomparison Project phase 5 (CMIP5) models and their accuracy evaluation between 1960 and 2005.

| CMIP5 | Institutions | Spatial Resolution | Simulated Periods | Projected Periods | Precipitation | | Temperature | |
|---|---|---|---|---|---|---|---|---|
| | | | | | *r* | Rmean | *r* | Rmean |
| CCSM4 | NCRA(America) | $1.25° \times 0.94°$ | 1850–2005 | 2006–2100 | 0.48 | 0.41 | 0.98 | −0.04 |
| CMCC-CM | CMCC(Italy) | $0.75° \times 0.75°$ | 1850–2005 | 2006–2100 | 0.13 | 0.38 | 0.91 | −0.10 |
| CNRM-CM5 | CNRM-CERFACS(France) | $1.40° \times 1.40°$ | 1850–2005 | 2006–2100 | 0.40 | 0.12 | 0.98 | −0.09 |
| HadGEM2-ES | MOHC(Britain) | $1.875° \times 1.25°$ | 1859–2005 | 2006–2099 | 0.52 | −0.01 | 0.97 | −0.04 |
| MIROC4h | MIROC(Japan) | $0.56° \times 0.56°$ | 1850–2005 | 2006–2035 | 0.41 | 0.21 | 0.98 | 0.13 |
| MIROC5 | MIROC(Japan) | $1.40° \times 1.40°$ | 1850–2005 | 2006–2100 | 0.50 | 0.39 | 0.98 | 0.14 |

Through the accuracy test in the UMHRB, the temperature in all CMIP5 models shows better correlations between CMIP5 simulation data and ground observation data as the *r* of the temperature greater than 0.91 and the *Rmean* of annual temperature smaller than 0.14. However, the precipitation in four CMIP5 models, such as CNRM-CM5, HadGEM2-ES, MIROC4h, and MIROC5, shows relative better correlations between CMIP5 simulation data and ground observation data as the *r* of precipitation greater than 0.40 and *Rmean* of annual precipitation smaller than 0.39.

It should be noted that the MIROC4h has a short projected period (2006–2035) for the future. Finally, three relatively better CMIP5 models, such as CNR, Had, and MIR, have been chosen in this study.

The study would analyze the drought events from January 2020 to November 2099 under the representative concentration pathways of 4.5 (RCP4.5, medium concentration) and 8.5 (RCP8.5, high concentration). At the same time, the historical drought situation from 1960 to 2015 was also be analyzed for comparison.

## 2.2. Methods

### 2.2.1. The Standardized Precipitation Evapotranspiration Index (SPEI)

The SPEI is a simple multi-scale drought index that based on the climatic water balance combining the precipitation (P) and the potential evapotranspiration (PET).

Firstly, calculate the PET. The PET in the historical periods (1960–2015) were calculated by Penman-Monteith [50] equation because Penman-Monteith method was proposed as the first choice to obtain PET recommended by FAO [50] and the ground observed datasets have the needed variables.

While, the future PET from 2020 to 2099 were calculated by Thornthwaite [51] equation because the mean temperature was given by CMIP5 models, but some variables needed in the Penman-Monteith were lacking.

Secondly, calculate the difference between $P_i$ and $PET_i$ for the months $i$ according to:

$$D_i = P_i - PET_i \tag{1}$$

The calculated $D$ values are aggregated at different time scales:

$$D_n^k = \sum_{i=0}^{k-1} (P_i - PET_i), n \geq k \tag{2}$$

where $k$ (months) is the timescale of the aggregation and $n$ is the calculation month.

Thirdly, choose the probability density function of a three parameter log-logistic distributed variable to fit the datasets of $D_i$ values:

$$f(x) = \frac{\beta}{\alpha} \left(\frac{x-\gamma}{\alpha}\right)^{\beta-1} \left[1 + \left(\frac{x-\gamma}{\alpha}\right)^{\beta}\right]^{-2} \tag{3}$$

where $\alpha$, $\beta$, and $\gamma$ are scale, shape and origin parameters, respectively, for $D$ values in the range ($\gamma > D < \infty$). Parameters of the log-logistic distribution can be obtained following the L-moment procedure.

Lastly, the probability distribution function of the $D$ series, according to the log-logistic distribution, is given by

$$F(x) = \left[1 + \left(\frac{\alpha}{x-\gamma}\right)^{\beta}\right]^{-1} \tag{4}$$

With $F(x)$ the SPEI can easily be obtained as the standardized values of $F(x)$:

$$SPEI = W - \frac{C_0 + C_1 W + C_2 W^2}{1 + d_1 W + d_2 W^2 + d_3 W^3} \tag{5}$$

where

$$W = \sqrt{-2\ln(P)}, P \leq 0.5 \tag{6}$$

and $P$ is the probability of exceeding a determined $D$ value, $P = 1 - F(x)$. If $P > 0.5$, then $P$ is replaced by $1 - P$ and the sign of the resultant SPEI is reversed. The constants are $C_0 = 2.515517$, $C_1 = 0.802853$, $C_2 = 0.010328$, $d_1 = 1.432788$, $d_2 = 0.189269$, and $d_3 = 0.001308$.

The detailed descriptions of the SPEI calculation can be found in the references of Vicente–Serrano et al. [40] and Beguería et al. [52]. According to different SPEI values, drought categories divide into four grades, such as mild, moderate, severe and extreme drought (Table 2).

**Table 2.** Classification scale for the standardized precipitation evapotranspiration index (SPEI) and their drought categories.

| Drought Category | None ($D_0$) | Mild ($D_1$) | Moderate ($D_2$) | Severe ($D_3$) | Extreme ($D_4$) |
|---|---|---|---|---|---|
| SPEI Values | $-0.5 < $ SPEI | $-1.0 < $ SPEI $\leq -0.5$ | $-1.5 < $ SPEI $\leq -1.0$ | $-2.0 < $ SPEI $\leq -1.5$ | SPEI $\leq -2.0$ |

SPEI calculation can be divided into many timescales, such as one, three, six, nine, 12, and 18 months, and different timescales represent different water accumulation state. In this study, since the four distinct seasons in the Huai River Basin and the obvious seasonal variation characteristics of drought events, the timescales should not too short or too long. Therefore, the SPEI index of three-month time scale (SPEI-3) is suitable for identifying continuous drought events in the basin, because the three-month time scale would neither cause the change of drought index too fast nor too slow.

For the purpose of studying drought tendency and frequency in the year and four seasons, SPEI-12 in December of each year was used as the annual drought index indicating water shortage in a given year, and SPEI-3 in February, May, August, and November were used to indicate water shortage in a given winter, spring, summer, and autumn, respectively.

### 2.2.2. Mann-Kendall Trend Test

The non-parametric Mann-Kendall (M-K) test [53], recommended by the World Meteorological Organization, is applied in this study to detect the trends in climate variables and SPEI values to assess the quantification of the changes from the history to the future.

The non-normal distribution of atmospheric and hydrological variables makes the classical statistical methods invalid, however, the non-parametric test method is considered as a good tool for trend analysis. The M-K test has the advantage of the independent of data distribution and less sensitive to missing values and thus most commonly used for assessment of trends in different climate and hydrological variables [47].

Supposing one series $\{x_r\}$, $(r = 1, 2, 3, \ldots, n)$, the M-K test statistic ($S$) is estimated as follows:

$$S = \sum_{k=1}^{n-1} m_i \sum_{i=k+1}^{n} sign(x_i - x_k) \tag{7}$$

where:

$$sign(x_i - x_k) = \begin{cases} +1 & if \ (x_i - x_k) > 0 \\ 0 & if \ (x_i - x_k) = 0 \\ -1 & if \ (x_i - x_k) < 0 \end{cases} \tag{8}$$

The variance of $S$ [$Var(S)$] is estimated using Z statistics to determine trend significance:

$$Z = \begin{cases} \frac{S-1}{\sqrt{Var(S)}} & if \ S > 0 \\ 0 & if \ S = 0 \\ \frac{S+1}{\sqrt{Var(S)}} & if \ S < 0 \end{cases} \tag{9}$$

At the significant level of 0.10, 0.05, 0.01, and 0.001, the null hypothesis of no trend is rejected if the absolute value of Z is greater than 1.645, 1.960, 2.576, and 3.292, respectively. If Z is positive, the data sequence shows an upward trend; otherwise, if Z is negative, it shows a downward trend.

### 2.2.3. Identification of Drought Characteristics with Run Theory

Combining the obtained SPEI and the run theory proposed by Yevjevich [54], the drought events and relevant characteristic variables can be identified by Figure 2. The cumulative SPEIs during the drought duration (*DD*) is used to measure the magnitude of a drought event and called the drought severity (*DS*). The *DS* is calculated as follows [26]:

$$DS = \left| \sum_{i=1}^{DD} SPEI_i \right| \tag{10}$$

where: The *DD* used in this paper is the drought months in one year.

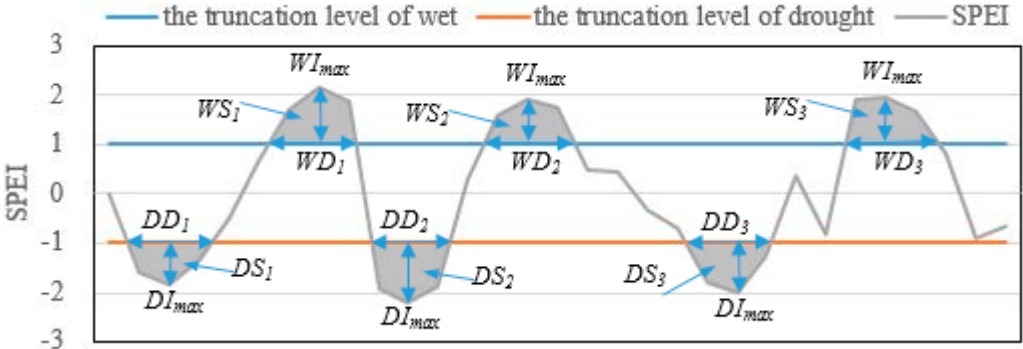

**Figure 2.** Drought or wet events and relevant characteristic variables by the run theory.

Each drought is characterized by two main properties: Drought duration and drought severity. In addition, the drought intensity (*DI*) is calculated as the ratio of drought severity and drought duration [26] and reflected the average water shortage during the drought duration. The peak value of drought severity, or called the max drought intensity (*DImax*), is obtained by the minimum value of *SPEI* value in a drought event at the special timescale and it reflected the most serious water shortage in the drought duration.

In this paper, the monthly time series of SPEI-3 was used in the Section 3.4 to discuss the changes of drought characteristics historically and in the future. Considering the time scale characteristics and the categories of *SPEI* index, this paper regards −1.0 as the truncation level of drought event, which can express the serious situation of drought.

In addition, in order to analyze the effects of precipitation and temperature on drought and wet events and the severity in Section 3.5, the SPEI-3 less than −1.0 and SPEI-3 greater than 1.0 for each month in each season are considered as seasonal severe drought and wetness condition, respectively. The wet severity (*WS*) during the wet duration (*WD*) calculate based on the equation:

$$WS = \sum_{i=1}^{WD} SPEI_i \qquad (11)$$

## 3. Results and Discussion

### 3.1. Characteristics of Precipitation and Temperature in the Basin

Figure 3 shows the time variations of the average annual precipitation and temperature historically and in the future, which were calculated from the ground observed data by the Thiessen polygons method in 1960–2015 and the CMIP5 models projected raster data by the arithmetic average method in 2020–2099, respectively. Considering the paper length, the maps of seasonal precipitation and temperature in four seasons were no longer display and only showed the statistical results in Tables 3 and 4, respectively.

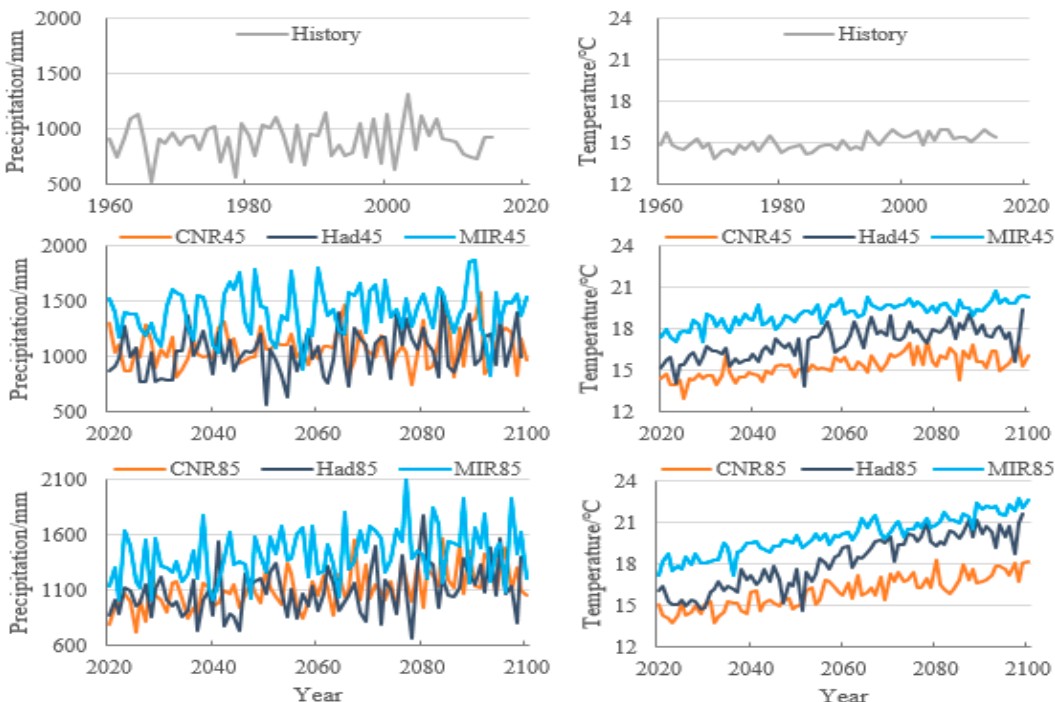

**Figure 3.** Changes of annual precipitation and temperature historically based on ground observed data and in the future based on the projected data under RCP4.5 (45) and RCP8.5 (85) scenarios.

**Table 3.** The mean value of annual and seasonal precipitation ($\overline{P}$/mm) and Mann-Kendall test values (Z) historically (1960–2015) and in the future (2020–2099).

| Timescale | Value | History | CNR | Had | MIR |
|-----------|-------|---------|-----|-----|-----|
| spring | $\overline{P}$ | 193.0 | 369.2/398.4 | 339.9/257.6 | 475.4/490.7 |
| | M-K/Z | −0.32 | 1.25/2.27 * | 3.04 **/2.26 * | 0.40/0.87 |
| summer | $\overline{P}$ | 457.1 | 418.3/428.9 | 436.3/463.1 | 531.2/514.5 |
| | M-K/Z | 0.40 | −0.26/3.36 ** | 1.14/2.24 * | −0.62/0.58 |
| autumn | $\overline{P}$ | 183.4 | 158.1/146.7 | 148.6/246.6 | 283.7/290.6 |
| | M-K/Z | −1.22 | 0.40/0.62 | −0.57/0.24 | 2.75 **/4.05 *** |
| winter | $\overline{P}$ | 69.5 | 131.1/139.7 | 111.8/124.3 | 116.1/128.5 |
| | M-K/Z | 1.16 | 1.84/1.37 | 0.98/2.24 * | 1.81/2.28 * |
| annual | $\overline{P}$ | 902.5 | 1077.6/1113.5 | 1035.7/1091.0 | 1406.5/1424.5 |
| | M-K/Z | −0.15 | 0.93/4.46 *** | 2.60 **/2.90 ** | 1.24/3.03 ** |

Notes: (1) The value on the left of sign "/" represents RCP4.5 scenarios, while the right represents RCP8.5 scenarios; (2) Values marked with +, *, **, and *** indicate that the series of SPEI values has been passed confidence test with the significant level of 0.10, 0.05, 0.01, and 0.001, respectively.

**Table 4.** The mean value of annual and seasonal average temperature ($\overline{T}$/°C) and Mann-Kendall test values (Z) historically (1960–2015) and in the future (2020–2099).

| Timescale | Value | History | CNR | Had | MIR |
|-----------|-------|---------|-----|-----|-----|
| spring | $\overline{T}$ | 15.09 | 14.69/15.29 | 17.04/12.31 | 19.70/20.42 |
|  | M-K/Z | 4.12 *** | 4.66 ***/6.46 *** | 4.23 ***/7.24 *** | 5.47 ***/9.18 *** |
| summer | $\overline{T}$ | 26.49 | 27.27/27.83 | 29.36/28.39 | 30.59/31.46 |
|  | M-K/Z | −0.29 | 4.05 ***/5.29 *** | 5.32 ***/9.56 *** | 7.63 ***/8.54 *** |
| autumn | $\overline{T}$ | 15.82 | 15.79/16.76 | 17.89/25.52 | 19.92/20.92 |
|  | M-K/Z | 3.58 *** | 6.40 ***/8.32 *** | 7.39 ***/8.60 *** | 6.18 ***/9.73 *** |
| winter | $\overline{T}$ | 2.73 | 3.42/4.05 | 3.86/6.33 | 6.30/7.49 |
|  | M-K/Z | 3.22 ** | 4.20 ***/5.91 *** | 5.69 ***/8.83 *** | 4.88 ***/7.67 *** |
| annual | $\overline{T}$ | 15.04 | 15.30/15.99 | 17.05/18.14 | 19.13/22.61 |
|  | M-K/Z | 4.16 *** | 7.43 ***/8.95 *** | 7.22 ***/9.90 *** | 8.25 ***/10.71 *** |

Notes: Symbols in the table have the same meaning as those in Table 3.

### 3.1.1. Changes of Annual and Seasonal Precipitation Historically and in the Future

In the annual and seasonal precipitation amount of the three models, the MIR values are the largest, while the CNR and Had values are smaller but a litter closer.

In terms of annual precipitation, the future mean values in the two emission scenarios (RCP4.5 and RCP8.5) of the three CMIP5 models are different, but they are all larger than the historical mean value (902.5 mm). In terms of seasonal precipitation, the projected results of MIR in four seasons under the two emission scenarios are larger than the historical levels. The projected seasonal precipitation of the other two CMIP5 models are also greater than the historical levels in most cases, expect for the CNR during the summer and autumn under RCP4.5 and RCP8.5 and the Had during summer and autumn under RCP4.5.

The three models showed an increasing trend for the annual precipitation in RCP4.5 and RCP8.5 scenarios. However, in RCP4.5 scenario only Had showed a significant increase trend with the significant level of 0.01. In RCP8.5 scenario, all of them passed the significant test with the significant level of 0.01.

For the trend of seasonal precipitation, the summer of CNR, autumn of Had, and summer of MIR under RCP4.5 scenario have shown an insignificant trend of decline. The other seasons or emission scenarios are showing an increasing trend but not all of them are significant. In RCP4.5 scenario, only Had's spring and MIR's autumn have passed the significant test with the significant level of 0.01. In RCP8.5 scenarios, there are more projected datasets pass the significant tests with the significant level of 0.01 or 0.05, such as the spring and summer of CNR, the spring, summer, and winter of Had, and the autumn and winter of MIR.

The differences of annual and seasonal projected precipitation of the three models may related to the different parameter setting and environmental variables selections [27]. Comparing the two emission scenarios, the amount, and the trend of precipitation in the RCP8.5 scenario is larger than that in RCP4.5 scenario in most cases. The emission of greenhouse gases has certain impacts on the precipitation, and the future projected precipitation increases slightly larger under high concentration emissions.

### 3.1.2. Changes of Annual and Seasonal Temperature Historically and in the Future

Under the same emission scenario in the annual and four seasons, the temperature projected by the three CMIP5 models are different. The order of values from large to small is MIR, Had, and CNR.

The future annual and seasonal temperature of MIR in the two emission scenarios (RCP4.5 and RCP8.5) are higher than the historical levels. The projected temperature of Had, expect for the spring (12.31 °C) in RCP8.5 scenarios bellowing the historical level (15.09 °C), are higher than the historical

levels. The projected temperature of CNR are higher than the historical levels in most cases expect for spring (14.69 °C) and autumn (15.79 °C) in RCP4.5 scenarios bellowing the historical levels of 15.09 °C and 15.82 °C, respectively.

The average annual and seasonal temperature of RCP8.5 emission scenario is higher than that of RCP4.5 emission scenario except for Had in spring and summer. The increase of greenhouse gas concentration leads to the increase of temperature in the most periods.

The annual and seasonal average temperature would all show a warming trend in the future. The M-K test (Z values) under RCP4.5 and RCP8.5 emission scenarios are greater than 4.00, all showing a strong warming trend.

The significant warming trend of the UMHRB in the future is consistent with the conclusion pointed out by Liu et al. [18] describing that the temperature warming continues to rise at a high level. With the increase of temperature, the water loss caused by evapotranspiration would increase.

From the above description and discussion of the changing characteristics of precipitation and temperature historically and in the future, the change trends of the two important climate variables and their significance are not keeping the same level, especially the temperature is showing a greater warming trend and stronger significance. Many previous studies have found that temperature variability plays an important role on characterizing droughts for the sensitivity of responses of arid/humid patterns change [24] and the largest increase of drought duration and intensity [20]. The existing fact that the water balance of the basin would different in the future could trigger the changes of future drought conditions and should pay broad and close attention and deeply research.

### 3.2. Drought Tendency Changes Historically and in the Future

3.2.1. The Interdecadal Linear Trend for the Monthly SPEI-3 Historically and in the Future

Time series of monthly SPEI-3 historically (1960–2015) and in the future (2020–2099) shown in the Figure 4 and their interdecadal linear trends were given in Table 5. The SPEI-3 sequence shows alternate wetting and drying changes and the trends of interdecadal drought or wet varies in different degrees in historical and future periods.

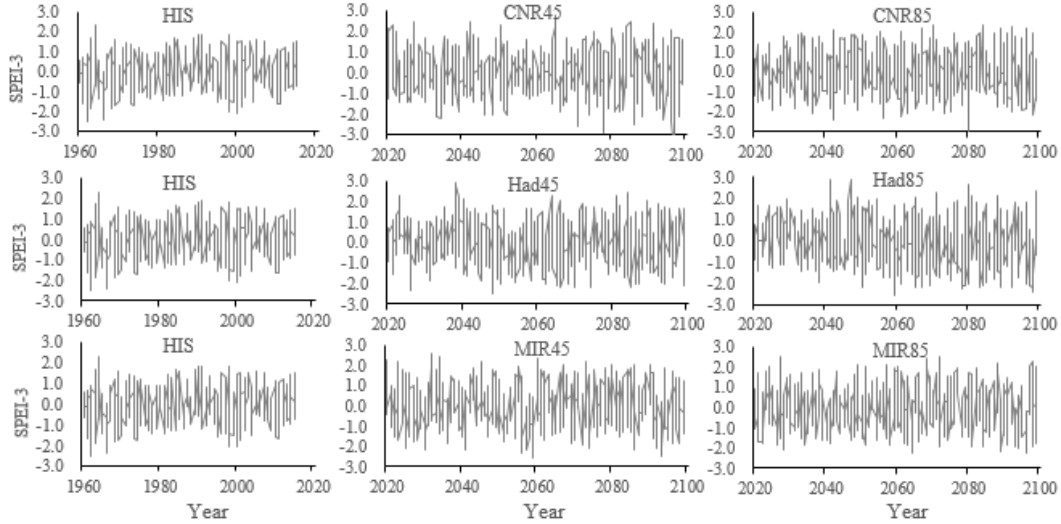

**Figure 4.** Changes of monthly SPEI index of three-month time scale (SPEI-3) historically (HIS) based on ground observed data and in the future based on the three CMIP5 models projected data under RCP4.5 (45) and RCP8.5 (85) emission scenarios.

**Table 5.** The interdecadal linear slope for the monthly SPEI-3 historically and in the future.

| Time Series | 1960s | 1970s | 1980s | 1990s | 2000s | 2010s | | Total | |
|---|---|---|---|---|---|---|---|---|---|
| History | 0.0001 | −0.0017 | 0.0007 | −0.0050 | 0.0012 | 0.0020 | | 0.0005 | |
| Time Series | 2020s | 2030s | 2040s | 2050s | 2060s | 2070s | 2080s | 2090s | Total |
| CNR45 | −0.0020 | −0.0019 | −0.0028 | −0.0022 | 0.0004 | −0.0028 | 0.0020 | −0.0059 | −0.0002 |
| Had45 | −0.0028 | 0.0095 | 0.0002 | 0.0055 | 0.0018 | 0.0023 | −0.0009 | 0.0031 | −0.0001 |
| MIR45 | −0.0059 | 0.0037 | 0.0077 | −0.0046 | −0.0021 | −0.0034 | 0.0017 | −0.0008 | −0.0001 |
| CNR85 | 0.0033 | −0.0002 | 0.0051 | −0.0070 | 0.0056 | 0.0053 | 0.0027 | −0.0014 | 0.0001 |
| Had85 | 0.0043 | −0.0041 | 0.0053 | −0.0056 | −0.0030 | −0.0060 | −0.0076 | −0.0033 | −0.0006 |
| MIR85 | 0.0029 | 0.0017 | 0.0062 | 0.0036 | 0.0039 | −0.0008 | −0.0022 | −0.0003 | −0.0002 |

Historical SPEI-3 values changed alternately in wetting-drying-wetting-drying from 1960s to 1990s, but it tended to be wetting in the early 21st century. The overall linear trend historically was rising and showing a trend of wetting.

Under RCP4.5 scenario, CNR shows a trend of drying in 2020s–2050s and wetting-drying alternation in 2060s–2090s. Both Had and MIR shows drying-wetting-wetting trend in 2020s–2040s, and their trends are consistent but different in magnitude. Since 2050s, the trends of Had and MIR are opposite, i.e., once Had is positive while MIR is negative during the same decade. The overall linear trend of the three CMIP5 models under the RCP4.5 scenario shows the trend of drought, especially for CNR that is slightly stronger in the future changes.

Under RCP8.5 scenario, CNR and Had shows wetting-drying-wetting-drying alternation through 2020s–2050s. However, CNR shows wetting at 2060s–2080s and drying at 2090s, while Had maintains drying at 2060s–2090s. MIR maintains the trend of wetting from 2020s to 2060s but altered drought trend from 2070s to 2090s. The overall linear trend of Had and MIR under RCP8.5 scenario present drying trend in the future and the drying trend of Had is much stronger. However, the CNR85 shows a relatively wetting trend on the contrary.

From the above analysis, it could be formed a conclusion that the drought trend of RCP8.5 scenario would be slightly stronger than that of RCP4.5 scenario. In the two emission scenarios of the above three CMIP5 models, some interdecadal trends are opposite and some interdecadal trends are similar. It should be paid attention to consider the uncertainty of climate models on future projected and distinguish the effects of different models in the further drought research. Overall, there would be a trend of drought in the basin under climate change and the drought trend would be stronger under high greenhouse gas emission which is mainly related to the significant warming increase of temperature.

### 3.2.2. Drought Tendency for Annual and Seasonal Timescales Historically and in the Future

Annual and seasonal drought situations in the 1960–2015 and 2020–2099 have shown in Figure 5. The SPEI values show periodic fluctuations, indicating historical and future wet and drought conditions. The trend changes of historical period and future period on annual and seasonal scales indicate the development characteristics of drought and wet conditions on corresponding time scales.

For the further trend tested with Mann-Kendall test (Table 6), the annual SPEI-12 during 1960–2015 showed an increasing trend which indicated a trend of wetting in the watershed. However, in the future, all models and scenarios except CNR85 is displaying a trend of drought, especially Had85 have passed the confidence test at the significant level of 0.001. In spring, the trend historically was almost unchanged. However, in the future, some models and scenarios, i.e., CNR45, Had45, and CNR85, show the trend of wetting while others, i.e., MIR45, Had85, and MIR85, show the trend of drought. However, they all have not passed the confidence test at the significant level of 0.05. Summer SPEI-3 historically showed a trend of wetting and through the confidence test at the significant level of 0.05. In the future, except for CNR85 showing a trend of wetting, the other models and scenarios towards to drought, especially MIR45 and MIR85 passing the confidence test at the significant level of 0.01 and 0.05, respectively. In autumn, there was a slight trend of drought historically. While, except for MIR45 and MIR85 showing a trend of wetting, there is still a trend of drought in the future, especially for the

Had45, CNR85, and Had85 which have passed the confidence test at the significant level of 0.01, 0.05, and 0.001, respectively. In winter, it showed a wetter trend in both the history and the future, but the future trend is slighter than the history.

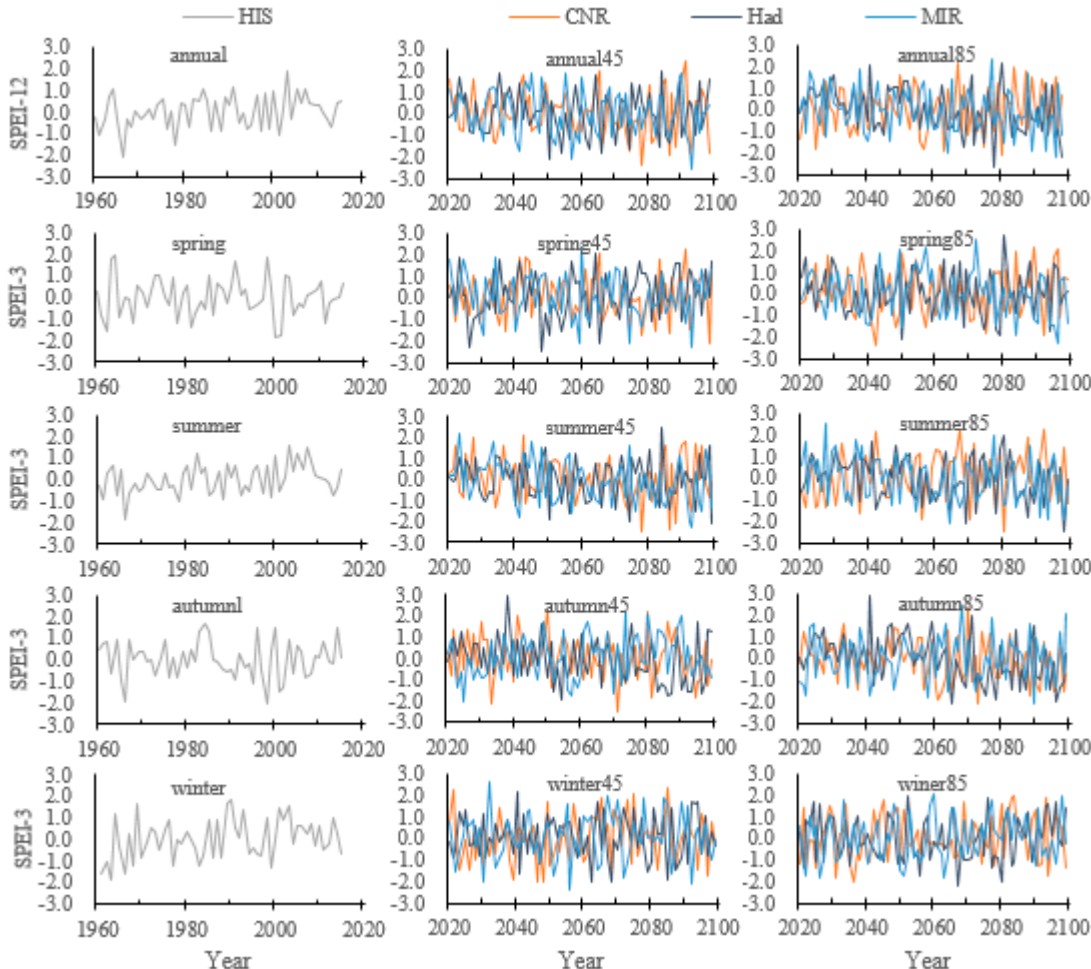

**Figure 5.** Changes of annual (SPEI-12) and seasonal (SPEI-3) drought indices historically and in the future.

**Table 6.** The Mann-Kendall test for annual (SPEI-12) and seasonal (SPEI-3) drought indices historically and in the future.

| Timescales | History | CNR45 | Had45 | MIR45 | CNR85 | Had85 | MIR85 |
|---|---|---|---|---|---|---|---|
| annual | 1.56 | −0.75 | −0.04 | −1.21 | 1.16 | −3.30 *** | −1.77 + |
| spring | 0.01 | 0.36 | 1.81+ | −0.47 | 0.62 | −0.10 | −1.72 + |
| summer | 2.08 * | −1.00 | −0.50 | −2.65 ** | 1.76 + | −1.68 + | −2.20 * |
| autumn | −0.59 | −.43 | −2.70 ** | 1.47 | −2.28 * | −3.87 *** | 0.71 |
| winter | 1.89 + | 0.91 | 0.25 | 0.70 | 0.70 | 0.39 | 1.24 |

Notes: Values marked with +, *, **, and *** indicated that the series of SPEI values has been passed confidence test with the significant level of 0.10, 0.05, 0.01, and 0.001, respectively.

Historically periods during 1960–2015, the watershed climate and water status tends to be wetter except in autumn. In the future periods during 2020–2099, besides winter, the SPEI of other seasonal and annual timescales would decrease significantly, and the future climate and water status would have a certain degree of drought trend, especially in summer and autumn. The reasons explored were that: 1) winter temperature is low and so the potential evapotranspiration calculated by Thornthwaite method

may be lower [51]. Thus, the values of precipitation minus potential evapotranspiration may appear slightly larger and seems that more water storage exists in the basin. 2) Summer precipitation is more abundant but the temperature is relatively high and autumn is generally sunny, hot, and rainless in the basin. Meanwhile, the significant trend of temperature rise is greater than precipitation under RCP4.5 and RCP8.5 emission scenarios. Higher temperature calculates higher potential evapotranspiration through Thornthwaite method. Therefore, in the future drought prediction and assessment research, it should pay attention to the influence of temperature rise on evapotranspiration, and the potential evapotranspiration should not be neglected in drought analysis.

### 3.3. Drought Frequency Changes Historically and in the Future

#### 3.3.1. Interdecadal Variation and the Whole Periods of Drought Frequency for Monthly SPEI-3 historically and in the Future

Figure 6 illustrates that the frequency of mild drought in interdecadal scale varies from 10% to 20% historically and between 8% and 23% in the future under RCP8.5 scenario. However, the frequency of mild drought under RCP4.5 scenario is showing a downward trend and approaching to 10% in 2090s. The moderate drought frequency between 1960s and 2010s was showing a downward trend from 10% to 5%, while that increase slightly under RCP4.5 and shows periodic fluctuations under RCP8.5 with the frequency both increasing to more than 5%. Severe and extreme drought frequency were showing the highest in 1960s and then approaching to zero in 1970s–1980s. Although they slightly increased after 1980s, the drought frequencies were almost showing a decline as a whole. In the future, the frequency would increase significantly compared with the beginning of the 21st century, especially at the end of the 21st century.

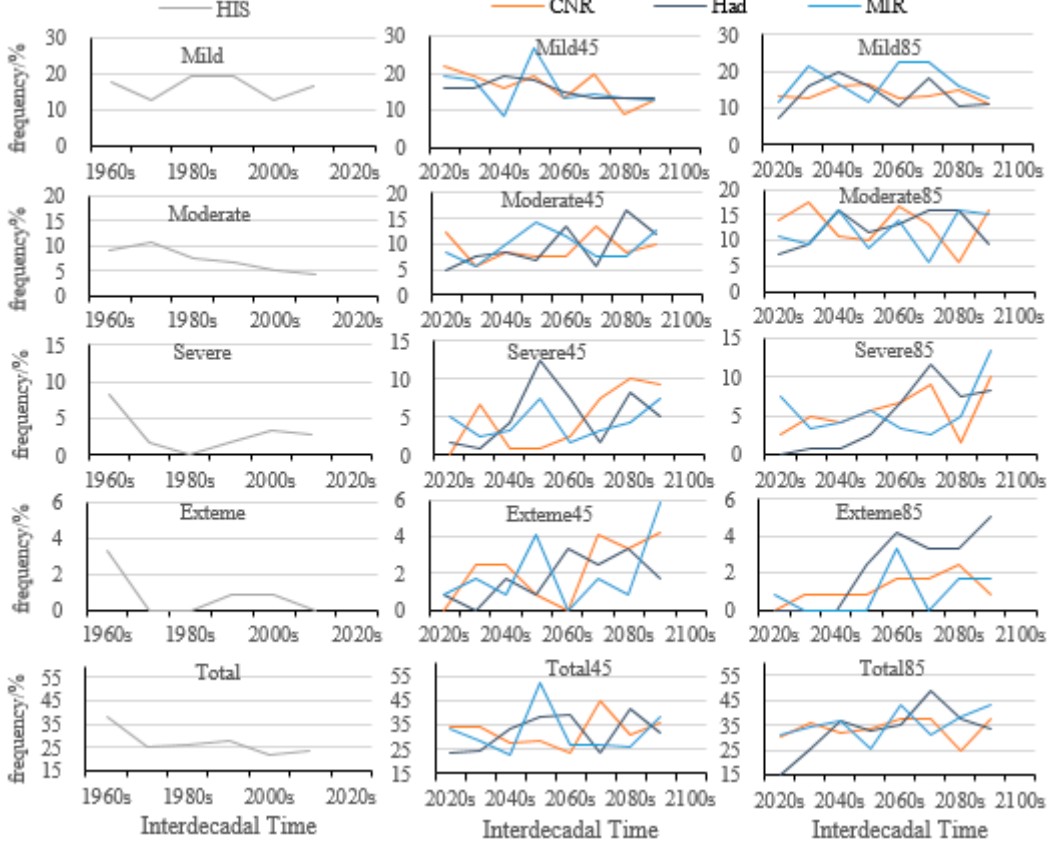

**Figure 6.** Interdecadal variation of drought frequency (%) at different drought categories historically (1960–2015) and in the future (2020–2099).

The overall performance of historical total drought (SPEI ≤ −0.5) frequency was showing a downward trend, in which it showed more than 35% in 1960s, and declined more than 25% in 1970s–1990s and then dropped to less than 25% when entering the 21st century. After the 2020s, the total drought frequency of RCP4.5 scenario would increase slightly compared with the beginning of the 21st century and return to more than 35% at the end of the 21st century. While, the changes of drought frequency under RCP8.5 scenario is more obvious for the increasing trend and magnitude.

In the whole periods, Table 7 shows that mild drought had a higher frequency historically than that in the future, except for CNR45 and MIR85. The frequencies of moderate, severe, extreme, and total drought in the future periods during 2020–2099 were higher than that historically during 1960–2015, especially under RCP8.5 emission scenario. To sum up, we can draw a conclusion that the frequency of drought above moderate categories may increase in the future and the frequency of total drought (SPEI ≤ −0.5) under RCP8.5 scenario is higher than that under RCP4.5 scenario.

**Table 7.** Drought frequency (%) in different drought categories in the whole history (1960–2015) and the long future (2020–2099).

| Drought Category | History | CNR45 | Had45 | MIR45 | CNR85 | Had85 | MIR85 |
| --- | --- | --- | --- | --- | --- | --- | --- |
| Mild | 16.22 | 16.37 | 15.54 | 15.75 | 13.76 | 13.76 | 16.89 |
| Moderate | 7.44 | 9.18 | 9.38 | 9.70 | 13.03 | 12.30 | 11.89 |
| Severe | 2.98 | 4.69 | 5.21 | 4.38 | 5.63 | 4.80 | 5.63 |
| Extreme | 0.89 | 2.19 | 1.77 | 1.98 | 1.15 | 2.29 | 0.94 |
| Total | 27.53 | 32.43 | 31.91 | 31.80 | 33.58 | 33.16 | 35.35 |

The increase of drought frequency in the future indicates that even with the increase of precipitation as mentioned in Section 3.1, the significant temperature rise plays a bigger role on the drought conditions and this performance is more obvious under RCP8.5. Sheffield and Wood have also proposed the same conclusion that the increases in precipitation are offset by increased evaporation, which was closely related to temperature [16].

### 3.3.2. Drought Frequency of Annual and Seasonal SPEI Historically and in the Future

Annual and seasonal drought frequencies in different drought categories shown in Figure 7. In terms of annual series, the frequencies of mild and extreme drought would increase or decrease under different future models. However, the frequencies of moderate, severe and total drought of the most models would increase in the future, except that MIR45 of severe drought would be decreased. In spring, the frequencies of mild drought (except MIR85), moderate drought (except MIR45), extreme, and total drought increase as a whole in the future. However, the frequency of severe drought in the future either increase or decrease. In summer, moderate, severe, extreme, and total drought would increase in the future than historically. The difference between the history and the future is obvious, which is consistent with the trend of Mann-Kendall test results. The effect of high temperature on water loss in summer is greater than that of precipitation. In autumn, mild drought except for Had45 and extreme drought for all CMIP5 models and scenarios had a higher frequency historically but a lower frequency in the future. However, the frequencies of moderate, severe and total drought would enlarge in the future. In winter, the frequency of mild drought, except for Had45 and MIR85, decreases in the future. While, the frequency of moderate drought, except for Had45 and MIR45, shows higher in the future. Severe drought in winter would reduce in the future, while extreme drought and total drought would increase in the future.

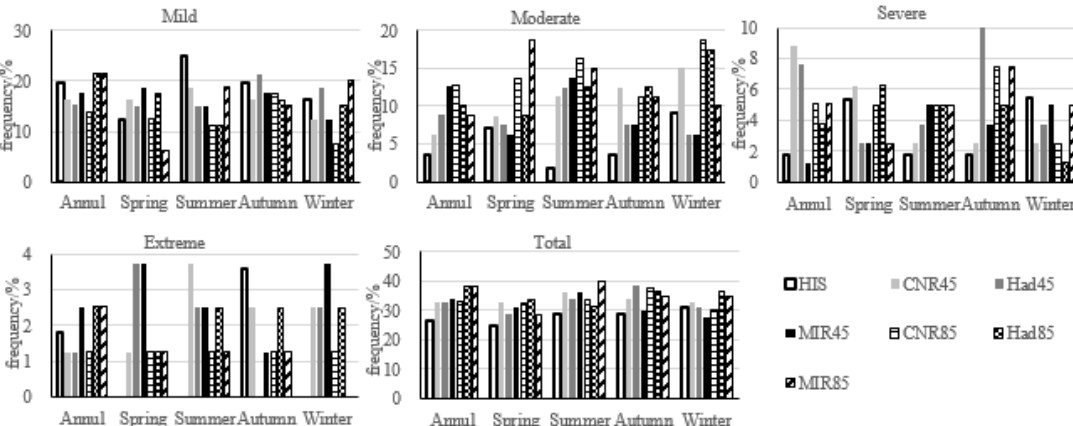

**Figure 7.** Annual and seasonal drought frequencies of different drought categories historically and in the future.

Overall, except in winter, the frequency of total drought in annual and other seasons is more than 30% in the future, which is greater than that historically with the value below 30%. Extreme drought would occur in spring, summer, and winter in the future, which never happened historically. Extreme drought in autumn would decrease. Moderate and severe drought in summer and autumn increase significantly under all models and scenarios, while moderate drought in spring and winter would increase but severe drought in spring and winter would decreases under the most models and scenarios. Mild drought in spring would increase but in the other seasons would decrease under the most models and scenarios.

The results that the UMHRB would suffer an increasing drought frequency in the future is consistent with Li et al. [26]. In other parts of the world or China [18–20], the frequency of droughts have been found to increase with the increase in temperature and changes in precipitation patterns.

### 3.4. Drought Characteristics Changes Historically and in the Future

### 3.4.1. Annual Drought Characteristics Changes Historically and in the Future

Figure 8 shows the annual time series for different annual drought characteristic values, such as drought severity, drought months, drought intensity, and the max drought intensity. The above values were statistics when the monthly SPEI-3 below the drought threshold level of −1.0 in each year from January to December.

For indicating the trend of the time series of different annual drought characteristic values historically and in the future, Table 8 presents the linear slope values of them. It could be found that the four annual drought characteristic values showed a downward trend historically. The drought severity and drought months peaked in 1966, 1978, and 2001, but the peak value continued to decline. While, the four annual drought characteristic values in the future under RCP4.5 and 8.5 scenarios showed an upward trend, but the performances were different in different models and scenarios. Under the RCP4.5 emission scenario, the trend of annual drought characteristic values changes from largest to smallest in the range of CNR45 > Had45 > MIR45. Under the RCP8.5 emission scenario, the trend of drought intensity from largest to smallest is CNR85 > MIR85, however, the others are Had85 > MIR85 > CNR85. In terms of the magnitude of changes in the four drought characteristic values, CNR45 and Had85 have the largest increase but MIR45 has the lowest increase. In addition, MIR45 maintains its original state on the annual max drought intensity.

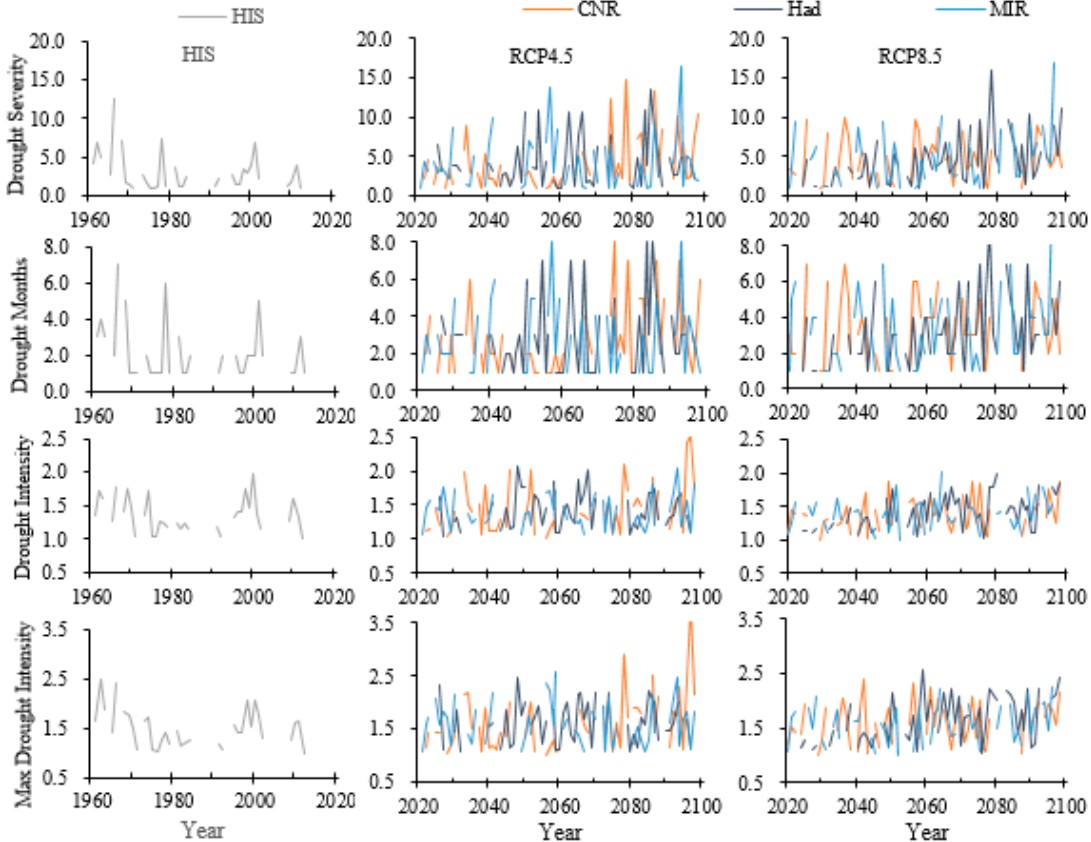

**Figure 8.** Changes of annual drought characteristic values historically (1961–2015) and in the future (2020–2098).

**Table 8.** The linear slope of annual drought characteristic values historically and in the future.

| Variations | History | CNR45 | Had45 | MIR45 | CNR85 | Had85 | MIR85 |
|---|---|---|---|---|---|---|---|
| drought severity | −0.0456 | 0.0587 | 0.0187 | 0.0092 | 0.0113 | 0.0683 | 0.0301 |
| drought months | −0.0266 | 0.0287 | 0.0113 | 0.0028 | 0.0021 | 0.0356 | 0.0164 |
| drought intensity | −0.0012 | 0.0053 | 0.0006 | 0.0006 | 0.0023 | 0.0053 | 0.0019 |
| max drought intensity | −0.0051 | 0.0099 | 0.0016 | 0.0000 | 0.0030 | 0.0097 | 0.0032 |

These results are closer to those of Li et al. [26] whom also found that drought risks including duration, severity, and intensity during 1961–2010 showed a decreasing trend in the same area while drought is expected to rise in duration, severity, and intensity from 2010–2099 under the RCP8.5 scenario of the HadGEM2-ES climate model.

3.4.2. Top Values of Annual Drought Characteristics Historically and in the Future

Combining the Figures 4 and 8, the top five drought characteristic values historically and in the future are further counted in Table 9. It can be seen that the two scenarios in the future overall reflect that the maximum drought characteristics are more serious than the historical period. It indicated that the annual drought severity would be greater, the drought months would be increased, and the drought intensity and the max drought intensity would be enhanced. However, it also should be pointed out that Had45 and MIR85 are slightly lower than other models and scenarios in terms of the top five max drought intensity.

**Table 9.** Values of top five annual drought characteristics historically and in the future.

| Variations | History | CNR45 | Had45 | MIR45 | CNR85 | Had85 | MIR85 |
|---|---|---|---|---|---|---|---|
| drought severity | 12.40/1966 | 14.69/2078 | 13.50/2085 | 16.39/2093 | 10.84/2080 | 16.03/2078 | 16.85/2096 |
| | 7.40/1978 | 13.24/2086 | 11.00/2054 | 13.77/2057 | 9.83/2036 | 11.06/2098 | 10.16/2064 |
| | 7.11/1968 | 12.28/2074 | 10.89/2083 | 9.91/2041 | 9.69/2056 | 10.49/2089 | 9.40/2022 |
| | 6.83/2001 | 10.36/2098 | 10.60/2066 | 9.16/2086 | 9.64/2025 | 9.74/2075 | 9.35/2047 |
| | 6.83/1962 | 10.21/2092 | 10.60/2062 | 8.70/2030 | 9.31/2075 | 9.73/2083 | 9.09/2084 |
| drought months | 7/1966 | 8/2074 | 8/2085 | 8/2093 | 7/2036 | 9/2078 | 11/2096 |
| | 6/1978 | 7/2078 | 8/2083 | 8/2057 | 7/2025 | 7/2089 | 7/2047 |
| | 5/1968 | 7/2086 | 7/2054 | 6/2041 | 6/2056 | 7/2075 | 7/2084 |
| | 5/2001 | 7/2092 | 7/2066 | 5/2086 | 6/2091 | 7/2083 | 6/2022 |
| | 4/1962 | 6/2098 | 7/2062 | 5/2030 | 6/2063 | 6/2098 | 6/2040 |
| drought intensity | 1.99/2000 | 2.53/2097 | 2.07/2048 | 2.05/2093 | 2.17/2080 | 1.98/2080 | 2.03/2064 |
| | 1.77/1966 | 2.40/2096 | 2.01/2068 | 1.83/2086 | 1.89/2049 | 1.84/2098 | 1.81/2051 |
| | 1.76/1998 | 2.10/2078 | 1.87/2065 | 1.83/2098 | 1.87/2073 | 1.82/2092 | 1.79/2093 |
| | 1.74/1969 | 2.02/2052 | 1.85/2058 | 1.77/2027 | 1.87/2098 | 1.79/2067 | 1.78/2089 |
| | 1.73/1974 | 2.01/2046 | 1.76/2049 | 1.74/2030 | 1.86/2075 | 1.79/2096 | 1.68/2091 |
| max drought intensity | 2.50/1962.03 | 3.82/2097.01 | 2.46/2048.05 | 2.58/2059.06 | 2.94/2080.09 | 2.57/2059.04 | 2.27/2096.05 |
| | 2.40/1966.10 | 2.90/2078.07 | 2.31/2026.05 | 2.48/2093.07 | 2.41/2042.05 | 2.44/2098.08 | 2.24/2064.09 |
| | 2.08/2000.04 | 2.56/2071.11 | 2.22/2085.01 | 2.36/2056.02 | 2.34/2056.07 | 2.24/2078.07 | 2.15/2089.11 |
| | 2.07/1998.11 | 2.51/2086.09 | 2.18/2074.07 | 2.21/2057.12 | 2.27/2061.12 | 2.22/2092.07 | 2.08/2028.07 |
| | 1.87/1963.02 | 2.40/2096.12 | 2.18/2070.09 | 2.17/2041.09 | 2.14/2098.12 | 2.21/2067.02 | 1.99/2094.06 |

Note: The left of "/" was the value, while the right was the occurrence year or the year and month.

In terms of the periods of drought occurrence, the historical annual serious drought severity mainly occurred in the middle and late 20th century and the future serious drought severity would be happened in the middle and late 21st century. Nevertheless, there is a slightly difference that the drought severity performance after 2070s would be more prominent.

It also should be noted that the drought months in the future would be generally longer (basically more than six months) when the severe drought years occurred. Moreover, the drought intensity would be basically lower than the SPEI severe drought threshold of −1.50 given by the Table 2 and the maximum drought intensity would be close to or lower than the SPEI extreme drought threshold of −2.00. Thus, the severe and extreme drought characteristics of the basin would be more prominent under the future climate change, and it is necessary to strengthen the human response programs for the coming drought risk.

### 3.4.3. Multi-Year Average Values of Annual Drought Characteristics Historically and in the Future

Figure 9 showed the multi-year average characteristic values for drought conditions (SPEI-3 ≤ −1) historically (1961–2015) and in the future (2020–2098). It could be found that the multi-year average drought characteristics of the future are greater than that of the history and there would behave differently under the two scenarios in the future. In the same scenario of the three CMIP5 models, the difference between the multi-year average values of annual drought severity and drought months varies not very large, but the values under RCP8.5 scenario are larger than that under RCP4.5 scenario. The multi-year average values of drought intensity of RCP4.5 are higher than that of RCP8.5. Obviously, the reason is that the RCP8.5 have the larger drought severity but long drought months and so the ratio of drought severity to drought months of RCP8.5 is a litter smaller than that of RCP4.5. In terms of the multi-year average of max drought intensity, the three models under RCP4.5 scenario are higher than 1.592, while only two models of RCP8.5 are higher than 1.592 and the other is 1.577, but the Had85 has the largest number (1.646) under all scenarios.

The phenomenon of the different drought characteristics describing with different global climate models and emission scenarios is mentioned by the study of Vrochidou et al. [21]. Therefore, in the future research of the impacts of climate change on drought characteristics, attention should be paid to distinguish the results of different models.

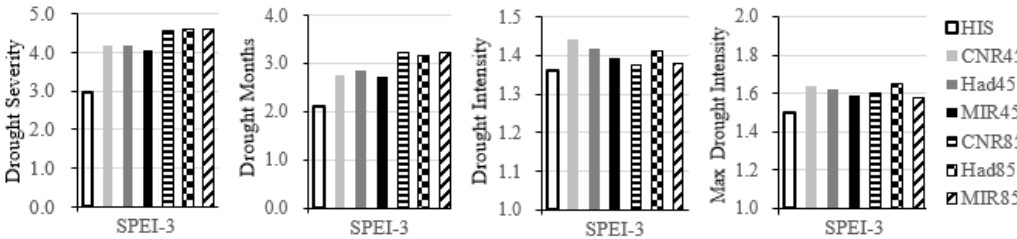

**Figure 9.** Comparison of multi-year average values of annual drought characteristic values for drought conditions (SPEI-3 ≤ −1) historically (1961–2015) and in the future (2020–2098).

### 3.5. Drought or Wet Response to Climate Change Scenarios

According to the principle of water balance, the input of precipitation and the output of evapotranspiration have a certain impact on the water storage in the basin. In order to analyze and discuss how the precipitation and temperature influence on the formation of seasonal drought and wet conditions historically and in the future on the basin, Figure 10 was drawn, and it chose the four seasons and divided them into drought and wet conditions. The situations that SEPI-3 is lower than −1.0 in all three months of each season is counted as drought condition, while the situations that SEPI-3 is higher than 1.0 in all three months of each season is counted as wet condition. Meanwhile, the seasonal precipitation and average temperature during the wet and drought condition were counted and shown in Figure 10.

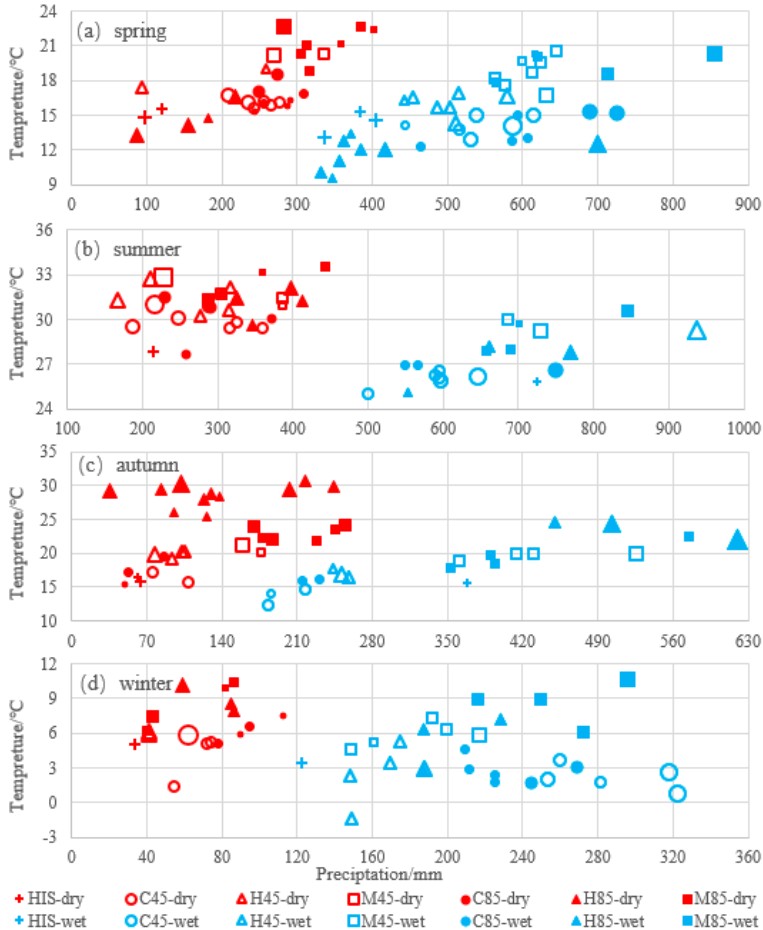

**Figure 10.** Distribution of seasonal precipitation and average temperature during seasonal severe drought and wet situation historically and in the future (The larger the blue and red symbols, the more the wet and drought severity they are, respectively.).

### 3.5.1. The Promotion of Precipitation and Temperature on the Formation of Severe Drought or Wet

The Figure 10 showed the historical and future precipitation and temperature distribution during drought and wet seasons. It could be found directly that the precipitation and temperature were the primary drivers of drought or wet situation. Less three-month precipitation and higher three-month average temperature promote the formation of severe drought situation, while higher precipitation and lower temperature are the conditions for the formation of severe wet situation.

The less precipitation and the higher temperature for the same model contribute to drought generation and make the drought more seriously. If the precipitation increased but the temperature was higher in the same period, the increasing evapotranspiration by the influences of temperature would also lead the basin turning to severe drought. In the case of less precipitation and lower temperature at the same time, it seems that the scarcity of precipitation may cause drought condition, but the water loss caused by low temperature may be relatively small, which made the drought severity sometimes may be lower.

In wet season, the more precipitation and the lower temperature could lead to wet situation. Obviously, that is because the input of water is greater than the water loss. Nevertheless, sometimes even if there is a lot of precipitation but with a higher temperature in the same period resulting in larger evapotranspiration output, the wet situation may be not necessarily greater than the situation of a slightly smaller precipitation with a lower temperature.

In view of the significant trend of temperature increase would be greater than that of precipitation under RCP4.5 and RCP8.5 scenarios, the future drought prediction and assessment should consider the changes of evapotranspiration caused by temperature rise and the potential evapotranspiration should not be neglected in drought analysis.

### 3.5.2. The Seasonal Drought Response to Climate Change Scenarios

According to the statistical results in Figure 10, the performance of successive drought or wet in the historical seasons was different from that in the future seasons with the three CMIP5 models and two emission scenarios. The precipitation and temperature in the future drought or wet seasons are statistically more than that in the historical drought or wet seasons, which corresponds to the previous conclusion in Section 3.1 that precipitation and temperature would increase in the future. However, the increase of precipitation does not mean that all droughts in the future season would be alleviated and the wet condition would be aggravated. According to the statistical results, sometimes the future drought severity even be more serious than the historical drought conditions, and the future wet severity may be weaker than the historical wet conditions. The main reason is that the coupling game between precipitation as input and temperature as output is the key to water storage in the basin, which has a great impact on the drought or wet situation. Considering the significance of temperature rise in the future is higher than that of precipitation increase, the impact of temperature could not be ignored even if precipitation increases. The drought severity in the future is not necessarily lower or even greater than that historically. In addition, the Figure 10 showed that the precipitation and temperature under RCP8.5 are generally slightly higher than that under RCP4.5, which brings about the distribution of drought or wet not inferior to RCP4.5 scenario.

In Figure 10, it also found that the drought or wet situation under the two scenarios of the three CMIP5 models are also more frequently than those historically in the statistical sense. These phenomena were corresponded to the conclusions from Section 3.2 to Section 3.4 that the future drought trend, frequency, and characteristics are still more serious. Finally, we can draw a clear conclusion that the increase of precipitation could not offset the impact of temperature rise for seasonal drought or seasonal wet and the future water storage with a higher frequency of drought situation would not optimistic in the basin. This conclusion corresponds to the study by Sheffield and Wood [16] whom mentioned that the increase in precipitation are offset by the increase evaporation in some regions.

In this study, the projected drought hazards are driven by an ensemble of three GCMs and two emission scenarios. The applicability accuracy of the CMIP5 model used in this basin is of statistical

significance, but the rough spatial resolution maybe bring uncertainty in future climate prediction and analysis. Future improvements in high resolution and parameter settings are needed to help improve the uncertainty of the analysis. However, greenhouse gas emissions would have a significant impact on climate change. The changes of precipitation and temperature have a great impact on the distribution of drought or wet situation in the basin. It needs to be continual widely concerned by the scientific and technological worker and the government to improve the response ability of drought risk under climate change and reduce the potential losses from drought.

## 4. Conclusions

In the upper and middle reaches of the Huai River Basin, the three CMIP5 models differ in describing the future seasonal and annual precipitation and temperature. The statistical values of MIROC5 are the largest in describing future annual and seasonal precipitation, while the values of CNRM-CM5 and HadGEM2-ES are smaller but closer. Meanwhile, MIROC5 are the biggest for describing the temperature in the future years and seasons, then the second is HadGEM2-ES and the third is CNRM-CM5.

Among the three CMIP5 models, the average precipitation and temperature in future years and four seasons would increase in most periods and scenarios, except that of CNRM-CM5 and HadGEM2-ES in some seasons and scenarios below the historical level. Under the three models, the annual precipitation would increase significantly on the significant level of 0.01 with the M-K test values greater than 2.90 in the future under high concentration emission scenarios. Most of the seasonal precipitation would increase but not in all four seasons. The annual and seasonal temperature of RCP4.5 and 8.5 scenarios under the three models showed an extremely significant warming trend on the significant level of 0.001 with the M-K test values greater than 4.00, especially under the high concentration path. The different changes of precipitation and temperature make the risk of future flood and drought greater.

The general trend of historical drought was wetting between 1960 and 2015, except the flat performance in spring and the arid showing in autumn. The historical frequency of mild drought ranged from 10% to 20% with interdecadal variation. The overall performance of historical total drought (SPEI ≤ −0.5) frequency showed more than 35% in 1960s and declined more than 25% in 1970s–1990s and then dropped to less than 25% when entering the 21st century. The frequency of historical moderate, severe and extreme drought and total drought and the historical annual characteristics of drought severity, drought months, and drought intensity all showed a downward trend. Among them, the historical drought severity and duration peaked in 1966, 1978, and 2001, but the peak value continued to decline.

Three CMIP5 models showed that there are differences in describing drought trends, frequency changes and drought characteristics in the future. Overall, the basin has a trend of drought in the future and the trend is stronger with a higher greenhouse gas emission concentration. The frequency of severe drought may increase in the future, which mainly related to the significant increase of temperature warming. The total drought frequency would increase slightly comparing with the beginning of the 21st century and return to more than 35% at the end of the 21st century, especially under RCP8.5 scenario. The drought characteristic values in the future show an upward trend but they are differences in the increase range under different models with RCP4.5 and 8.5 scenarios. On the whole, the drought characteristics of the basin in the future are more serious than the historical period, and the annual drought severity is greater, the drought duration is longer, the drought intensity and the maximum drought intensity are bigger. The maximum value of future annual drought severity and duration could reach 16.85 and 11 months under MIROC5 RCP8.5 scenarios, respectively, while, those of that were 12.40 and 7 months in the historical periods, respectively. In view of the serious and extreme drought characteristics of the basin under future climate change, it is necessary to strengthen the response research to drought risk.

Lower precipitation and higher temperature are the main factors contributing to the formation of drought events. Lower precipitation with lower temperature is generally weak in drought severity. If the precipitation increases but the temperature increases relatively larger, it should consider the change of evapotranspiration caused by the increase of temperature, which is more serious than that caused by the increase of precipitation but the small increase of temperature in the same period. Considering that the significant trend of precipitation increase in the future is lower than that of temperature rise, the influence of temperature should not be neglected in future drought research.

Drought under climate change needs continuous research and it should pay much attention to the precision test of climate model selection. The statistical downscaling or physical downscaling should be strengthened to obtain the higher spatial and temporal resolution values of climatic element for enhancing the credibility of drought research under the influence of climate change.

The results would be beneficial for the study of regional drought response to different climate models and emission scenarios and of great significance to regional water resources strategic planning, drought disaster prevention and control, and agricultural development planning.

**Author Contributions:** All authors contributed to the design and research of the manuscript. J.W. carried out the data analysis and prepared the draft of the manuscript. H.L. were responsible for the CMIP5 data collecting and analysis and made a lot of work for the compiling of tables and plotting of graphs. J.H., C.J., Y.X. and M.Z. participated in the research work and offered many important suggestions for the manuscript structure.

**Funding:** This research was supported by the Natural Science Foundation of the Jiangsu Higher Education Institutions of China (No.15KJB170019), the Priority Academic Program Development of Jiangsu Higher Education Institutions (PAPD), the Postdoctoral Research Foundation of Yangzhou University (No.137070375) and the Science and Technology Innovation Fund of Yangzhou University (No. 2016CXJ041).

**Acknowledgments:** The research data obtained by the CMA and WCRP and the authors would like to express much gratitude to the data provider.

**Conflicts of Interest:** The authors declare no conflict of interest.

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
