# Peer review of "Variations of Drought Tendency, Frequency, and Characteristics and Their Responses to Climate Change under CMIP5 RCP Scenarios in Huai River Basin, China"

_water, doi:10.3390/w11102174_

Round 1
Reviewer 1 Report
The research was done methodically correct. The description also does not raise major objections.
1) I am asking the authors to add a short overview of the climate models of Earth or Asia and explained why they used CMIP5 models for the analysis. It is increasingly said that models relying heavily on the effects of CO2 are not real.
2) I would like to ask to present in the table of results the analysis of the Pearson correlation coefficient (r ) for precipitation and temperature for other models, not only CNRM-CM5, HadGEM2-ES, MIROC5. This will justify choosing the above 3 models. Was the Pearson correlation coefficient (r ) statistically significant? At what level of p value?
3) The authors used the SPEI indicator. They explained very nicely why it is more valuable than, for example SPI, SWSI and how it counts. Please complete the SPEI calculation method with mathematical formulas. A scientific article on such a high level requires this.
4)There is no discussion of the results. A few references cited in the introduction can be used in the discussion.
Author Response
Thanks for the anonymous reviewer, the comments and suggestions. Those are beneficial for improving the quality of the manuscript. We have studied comments carefully and have made correction which we hope meet with approval. The main corrections in the manuscript and the responses to the reviewer’s comments are as following:
Point 1: I am asking the authors to add a short overview of the climate models of Earth or Asia and explained why they used CMIP5 models for the analysis. It is increasingly said that models relying heavily on the effects of CO2 are not real.
Response 1: The authors have added a short overview of the climate models of Global Climate Models (GCMs) and Regional Climate Models (RCMs), and explained why using CMIP5 models in the study. The revised parts have been highlighted with red colour in Paragraph 4 of Section 1.
Point 2: I would like to ask to present in the table of results the analysis of the Pearson correlation coefficient (r) for precipitation and temperature for other models, not only CNRM-CM5, HadGEM2-ES, MIROC5. This will justify choosing the above 3 models. Was the Pearson correlation coefficient (r) statistically significant? At what level of p value?
Response 2: Thanks for the reviewer's suggestions to present other models in Table 1. The authors had analysed six CMIP5 models in the previous studies before this study started, including the CCSM4, CMCC-CM, CNRM-CM5, HadGEM2-ES, MIROC4h and MIROC5. In the revised manuscript, the other models in Table 1 have been marked with red colour.
The Pearson correlation coefficient (r) in Table 1 were statistically significant and had passed the t test at P=0.05 level. A Description of the statistical results is supplemented in Section 2.1. Some corresponding sentences are rewritten in the revised manuscript.
Point 3: The authors used the SPEI indicator. They explained very nicely why it is more valuable than, for example SPI, SWSI and how it counts. Please complete the SPEI calculation method with mathematical formulas. A scientific article on such a high level requires this.
Response 3: The SPEI calculation method with mathematical formulas have been added in the Section 2.2.1. Thanks again for the reviewer's suggestions.
Point 4: There is no discussion of the results. A few references cited in the introduction can be used in the discussion.
Response 4: The manuscript has been supplemented and improved the discussion in Section 3. The results and discussions have been put together in this manuscript. Based on the results in the manuscript, some relevant studies are cited and discussed. The discussions are highlighted with red color in Section 3.
Thanks again for your good comments and suggestions.

Reviewer 2 Report
The manuscript is well prepared, presents an interesting results of the research and should be accepted.
The only comment: I recommend to include quantification of the results to the Conclusion section.
Author Response
Thanks for the anonymous reviewer, the comments and suggestions. These are beneficial for improving the quality of the manuscript.
Point 1: The manuscript is well prepared, presents an interesting results of the research and should be accepted. The only comment: I recommend to include quantification of the results to the Conclusion section.
Response 1: Thank you for your comments and appreciation of the manuscript. According to the comments, the authors have added some quantitative results to the Conclusion section, which have been highlighted with red colour.
